# MRI-Linac Economics II: Rationalizing Schedules

**DOI:** 10.3390/jcm11030869

**Published:** 2022-02-07

**Authors:** Peter A. S. Johnstone, John Kerstiens, Stuart Wasserman, Stephen A. Rosenberg

**Affiliations:** 1Department of Radiation Oncology, Moffitt Cancer Center, Tampa, FL 33612, USA; stuart.wasserman@moffitt.org (S.W.); stephen.rosenberg@moffitt.org (S.A.R.); 2Cancer Care Centers of Brevard, Melbourne, FL 32901, USA; john.r.kerstiens@gmail.com

**Keywords:** MR-guided radiotherapy, pancreas cancer, prostate cancer, health services research

## Abstract

Objective: Two benefits of MR-guided radiotherapy (MRgRT) are the ability to track target structures while treatment is being delivered and the ability to adapt plans daily for some lesions based on changing anatomy. These unique capacities come at two costs: increased capital for acquisition and greatly decreased workflow. An adaptive gated stereotactic body radiotherapy (MRgART) treatment routinely takes ~90 min to perform and requires the presence of both a physician and a physicist. This may significantly limit daily capacity. We previously described how “simple cases” were necessary for proton facilities to allow for debt management. In this manuscript, we seek to determine the optimal scheduling of different MRgRT plans to recoup capital costs. Materials/Methods: We assumed an MR-linac (MRL) was completely scheduled with patients over workdays of varying duration. Treatment times and reimbursement data from our facility for varying complexities of patients were extrapolated for varying numbers treated daily. We then derived the number of adaptive and non-adaptive patients required daily to optimize the schedules. HOPPS data were used to model reimbursement. Results: A single MRL treating 14 non-gated, non-adaptive IMRT patients over an 8 h workday would take about 4.8 years to cover initial acquisition and installation costs. However, such patients may be more quickly and efficiently treated with a conventional linear accelerator, while MRgART cases may only be treated with an MRL. By treating four of these daily, that same MRL room would cover costs in 2.4 years. Personnel, maintenance costs, and profit further complicate any business case for treating non-adaptive patients or for extending hours. Conclusions: In our previously published paper discussing proton therapy, we noted that debt is not variable with capacity; this remains true with MRgRT. Different from protons, a clinically optimal case load of adaptive patients provides an optimal business case as well. This requires a large patient cadre to ensure continuing throughput. As improvements in MRgRT are brought to the clinic, shorter adaptive and non-adaptive treatment times will help improve the timeframe to recoup costs but will require even more appropriate patients.

## 1. Introduction

Some years ago, we were concerned by the rush to proton therapy, funded in many cases by venture capital. Many pro formas in those days spoke to the potential return on investment of large centers treating predominantly prostate cancer. The potential number of cases was huge, throughput would not be an issue, and capital flowed into the first few centers. However, routinely using technology so scarce and expensive for cases without any survival benefit was rightly considered unreasonable in a cost-constrained health care environment [1,2]. Moreover, prostate cancer has such a high incidence that most standard RT practices cannot afford to give such cases up in the absence of a clear advantage in outcome. We wrote in 2012 that large proton centers should be expected to treat a mix of complex (slow, difficult) and simple (faster, easier) cases, or they would never be able to manage their debt [3]. Six years later, despite the shift to smaller, single room centers [4], many of the same points remain true.

We have noticed a similar phenomenon with MRgRT. This is a new technology where magnetic resonance imaging is used to perform both gating (delivery based on internal organ motion) and adaptive (generating a new treatment plan “on the fly”) treatments daily. These are usually delivered as brief courses of stereotactic body radiotherapy (SBRT). Both gated delivery and adaptive planning account for intra-fraction organ motion, but the latter potentially allows for an entire new plan to be generated and treated each day. While protons are more expensive, MRgART currently has a smaller and more discrete patient population appropriate for adaptation.

Finally, because of this complexity, MRgART requires both a physician present to recontour nearby organs at risk (OAR) daily and to perform the adaptive plan, and a physicist to ensure the safety of a newly generated plan. This complicates matters for a technology that costs several times more than a standard linear accelerator.

As MRgRT becomes more widely used, we wondered if there is a dichotomy between the best clinical use of daily gated adaptive planning and the use with best throughput: non-gated SBRT. In this manuscript, we sought to document an optimal relationship between the several potential clinical situations that will allow for the equipment to be used to its best potential while allowing it to be paid off reasonably quickly.

## 2. Methods

Primary in any such discussion of technology capacity are the rules of what is not included. In this case, daily treatment hours will not include the time required for machine warm-up and daily machine QA. Similarly, we removed simulations from the daily treatment times, recognizing that machine and therapist time must be added to the workday to include this function.

The following assumptions were used in our model:


Debt maintenance was the base calculation. Obviously, it would be more accurate to include personnel costs, overhead, and other operational costs, but those vary much more between sites.Analysis unit was per room, used 5 days weekly and 52 weeks annually.Given the capital cost, we optimized the schedule for patients for whom the discrete capabilities of this technology are most essential. There are limited opportunities for conventional IMRT patients to benefit from MRgRT (although exceptions exist such as hypofractionated-gated treatment in high-risk regions, but these are infrequent). Non-gated SBRT are often more easily and quickly performed on conventional linear accelerators but may be treated given MRL capacity or unique clinical situations. Finally, MRgART was favored over gated-only delivery given higher reimbursement.Capacity and time assumptions based on our experience (Table 1). We assumed adequate eligible patients to enable full scheduling at all times.Practically, a maximum of five MRgART daily has been our experience given the following limitations: lack of arc therapy, dose rate, and patient tolerability with gating.Hospital Outpatient Prospective Payment System (HOPPS) reimbursement at Florida rates as per Table 1.An acquisition cost of $9M was derived from web sources [5]. Personnel, operations, and maintenance costs and profit were not included.Charity cases are not modeled as this is a best-case scenario.


Optimal schedules were then formatted for 7, 8, 9, and 10 h days of treatments. In truth, adding two simulation slots daily is essential to continue uninterrupted treatments, and would be expected to make each day about an hour longer.

## 3. Results

Data are shown in Table 2. A single MRL treating non-gated, non-adaptive IMRT patients over an 8 h workday would take about 4.8 years to cover initial acquisition and installation costs. However, by including four gated MRgART cases daily, that same room would cover costs in 2.4 years. By converting therapists to 10 h shifts, personnel costs would be increased nearly proportionally to the increase in hours. Thus, these conclusions would extend to longer days.

## 4. Discussion

It must be noted that any estimate of the MRL breakeven costs with respect to earnings is unique to our situation. Others should expect variability as a function of workload, the execution times of the procedures, the personnel, and the presence of different times in their institutions.

As noted with proton therapy, we consider MRgRT to be a niche field. Few centers have the patient load requiring the more expensive and slower MRL technology, and we certainly expect few community centers to be able to make the financial calculus work. However, even for those centers that do: “although the treatment is based on the laws of physics, we ignore the laws of economics at our own peril” [6]. In this case, radiation therapy is like an airline: both require very expensive equipment that optimally generates revenue only when occupied. As the pandemic has proven, airplanes are nothing more than a drain on profitability when they are not flying. Similarly, an MRL with excess capacity is a very expensive magnet.

Many large academic centers will offer proton therapy, MRgRT, and other new technologies because they have access to cheaper cash (debt and donations) that make the pro formas slightly better. Such centers also have access to a larger pool of patients; thus, they may have more ‘eligible’ patients for these technologies. Even so, there must still be a mechanism that allows for paying back the initial outlay over some discrete and reasonable length of time. In our initial [3] and subsequent [4] analyses of proton therapy, the far larger cost required a significant portion of the schedule to be devoted to simple cases. That is not the case here since SBRT provides better reimbursement than conventional IMRT, but it does require a far larger pool of patients given the brief five-fraction course.

It must be noted that new clinical scenarios are being investigated to increase the pool of potentially adaptable patients for MRgART technology. For instance, retroperitoneal sarcomas being treated pre-operatively are the subject of clinical study in our center.

Finally, as with other scarce resources, charity care must be tightly controlled [7]; this allows revenue to be maximized. To that point, only one of our first 150 cases was a charity case.

## 5. Conclusions

Distinct from proton therapy, an optimal business model for MRgRT includes as many adaptive gated plans as is practical. This requires a large pool of patients since these cases are relatively rare; otherwise, the high capital cost and slow throughput are disadvantages compared with conventional linear accelerators. Upcoming improvements in MRgRT technology may improve treatment times and throughput over the next few years, which would affect this economic analysis. In any case, having sufficient appropriate patients is very important and excess capacity is discouraged.

## Figures and Tables

**Table 1 jcm-11-00869-t001:** HOPPS reimbursement values.

	Time to Deliver (min)	HOPPS Reimbursement
IMRT, conventional dose (e.g., bladder boost at 1.8 Gy daily)	30	$520
SBRT, no gating (e.g., prostate SBRT)	45	$1691
SBRT, gating (e.g., peripheral lung)	60	$1691
SBRT, gating + adaptive (e.g., pancreas)	90	$3207

Note: Motion management comes with professional but not technical reimbursement.

**Table 2 jcm-11-00869-t002:** Number of daily patients and daily HOPPS reimbursement according to treatment times.

Tx Type	7 h	7 h	7 h	8 h	9 h	10 h
IMRT	14					
SBRT		9		1	1	1
Gated SBRT			1		1	2
MRgART			4	5	5	5
Daily HOPPS	$7280	$15,219	$14,519	$17,726	$19,417	$21,108
Years to fund capital outlay	4.8	2.3	2.4	1.9	1.8	1.6

## Data Availability

Not applicable.

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
