# Peer review of "MRI-Linac Economics II: Rationalizing Schedules"

_jcm, 2022, doi:10.3390/jcm11030869_

Round 1

Reviewer 1 Report

My compliments to the authors because this work addresses a problem of high interest and highly topical, given that the important technological change in recent years has led to a significant increase in costs to ensure a high quality of care.

Although original and significant in content, the work refers to a single institution and a single work situation. It should be specified in discussion that the estimates of the breakeven costs with respect to the observed earnings refer to the reported contingent situation and that there could be a variability, linked to the workload, the execution times of the procedures, the personnel and the presence of different times in other institutions.

Another possible objection is linked to the terms of comparison of the method only towards protons, which are a very expensive technique in the initial investment, but certainly reserved for a much more selected subgroup of patients than MRI-Linac.
Finally, the true balance between costs and benefits of this equipment is made globally with the evaluation of the possible improvement of the results of disease control, healing and limitation of side effects, optimally obtained with the use of MRI-Linac. A calculation should be made that takes into account the savings in clinical interventions on patients treated with this machine.

Author Response

My compliments to the authors because this work addresses a problem of high interest and highly topical, given that the important technological change in recent years has led to a significant increase in costs to ensure a high quality of care.

Thanks for the kind words.

 Although original and significant in content, the work refers to a single institution and a single work situation. It should be specified in discussion that the estimates of the breakeven costs with respect to the observed earnings refer to the reported contingent situation and that there could be a variability, linked to the workload, the execution times of the procedures, the personnel and the presence of different times in other institutions.

We have added words in the Discussion section to this point.

Another possible objection is linked to the terms of comparison of the method only towards protons, which are a very expensive technique in the initial investment, but certainly reserved for a much more selected subgroup of patients than MRI-Linac.

A sentence to that point has been added.

Finally, the true balance between costs and benefits of this equipment is made globally with the evaluation of the possible improvement of the results of disease control, healing and limitation of side effects, optimally obtained with the use of MRI-Linac. A calculation should be made that takes into account the savings in clinical interventions on patients treated with this machine.

Such a calculation is not possible now, and may only be made with more data about the clinical benefits of adaptive radiotherapy.  This technique as only recently become available, with the advent of MRL.  It remains a future goal to document how MRgART may impact survival.

Reviewer 2 Report

This manuscript, entitled "MRI-Linac Economics II: Rationalising Schedules", discusses the financial benefits of MR-guided radiotherapy in the field of radiology. In doing so, the article briefly discusses the method of radiotherapy and describes the corresponding models of delivery as well as the corresponding financial costs. The authors also highlight the correlation between treatment time and the corresponding treatment costs. The authors describe the pure treatment costs in time. The term "HOPPS" could be explained in more detail. For a specific cost analysis, the personnel and consumption costs could also be included in the analysis. Furthermore, it would be interesting to know why only one case was a charity case. Otherwise, the article is well written and gives a good overview of the cost situation of MG-guided radiotherapy.  

Author Response

This manuscript, entitled "MRI-Linac Economics II: Rationalising Schedules", discusses the financial benefits of MR-guided radiotherapy in the field of radiology. In doing so, the article briefly discusses the method of radiotherapy and describes the corresponding models of delivery as well as the corresponding financial costs. The authors also highlight the correlation between treatment time and the corresponding treatment costs. The authors describe the pure treatment costs in time.

The term "HOPPS" could be explained in more detail.

We have done this.

For a specific cost analysis, the personnel and consumption costs could also be included in the analysis.

This actually renders the data less generalizable.  Please see the explanation in Methods section 1.

Furthermore, it would be interesting to know why only one case was a charity case.

See our explanation in the manuscript.

Otherwise, the article is well written and gives a good overview of the cost situation of MG-guided radiotherapy.

Thank you for the kind words.